# Two Complementary Perspectives to Continual Learning: Ask Not Only *What* to Optimize, But Also *How*

**Timm Hess**
KU Leuven

**Tinne Tuytelaars**
KU Leuven

**Gido M. van de Ven**
KU Leuven

## Abstract

Recent years have seen considerable progress in the continual training of deep neural networks, predominantly thanks to approaches that add replay or regularization terms to the loss function to approximate the joint loss over all tasks so far. However, we show that even with a perfect approximation to the joint loss, these approaches still suffer from temporary but substantial forgetting when starting to train on a new task. Motivated by this 'stability gap', we propose that continual learning strategies should focus not only on the optimization objective, but also on the way this objective is optimized. While there is some continual learning work that alters the optimization trajectory (e.g., using gradient projection techniques), this line of research is positioned as alternative to improving the optimization objective, while we argue it should be complementary. To evaluate the merits of our proposition, we plan to combine replay-approximated joint objectives with gradient projection-based optimization routines to test whether the addition of the latter provides benefits in terms of (1) alleviating the stability gap, (2) increasing the learning efficiency and (3) improving the final learning outcome.

## 1 INTRODUCTION

Learning continually from a stream of non-stationary data is challenging for deep neural networks. When these networks are trained on something new, their default behaviour is to quickly forget most of what was learned before (McCloskey and Cohen, 1989; Ratcliff, 1990). Considerable progress has been made in recent years towards overcoming such 'catastrophic forgetting', for a large part thanks to methods using replay (Robins, 1995; Rolnick et al., 2019) and regularization (Kirkpatrick et al., 2017; Li and Hoiem, 2017). These methods work by adding extra terms to the loss function, and they can be interpreted as attempts to approximate the joint loss over all tasks so far.

Recently a peculiar property of replay and regularization methods was pointed out. These approaches tend to suffer from substantial forgetting when starting to learn a new task, although this forgetting is often temporary and followed by a phase of performance recovery (De Lange et al., 2023). We postulate that avoiding this 'stability gap' is important, both because the transient drops in performance themselves can be problematic (e.g., for safety-critical applications) and because doing so might lead to more efficient and better performing algorithms, as constantly having to re-learn past tasks seems wasteful. However, in preliminary experiments we find that the stability gap cannot be overcome by merely improving replay or regularization. This motivates us to propose that, instead, continual learning needs an additional perspective: rather than focusing only on *what* to optimize (i.e., the optimization objective), the field should also think about *how* to optimize (i.e., the optimization trajectory).

There are existing works that explore modifying the optimization trajectory as a mechanism for continual learning, but so far this line of research has been positioned as alternative to improving the optimization objective. A prime example is Gradient Episodic Memory (GEM; Lopez-Paz and Ranzato, 2017), which alters the optimization process by projecting gradients to encourage parameter updates that do not strongly interfere with old tasks. Crucially, GEM applies this optimization routine to optimizing the loss on the new task, while our proposal is that such modified optimization routines should be used to optimize approximations of the joint loss. As a first evaluation of the merits of our proposition, in this pre-registered report we plan to use GEM's gradient projection-based optimization routine to instead optimize replay-approximated versions of the joint loss. Using both domain- and class-incremental learning benchmarks, we test whether this combined approach reduces the stability gap, and whether this in turn leads to higher learning efficiency and better final performance.

# 2 TWO PERSPECTIVES TO CONTINUAL LEARNING

In this section we use conceptual arguments and preliminary data to develop the proposition that continual learning should focus not only on *what* to optimize, but also on *how*. We first describe the current dominant approach to continual learning (subsection 2.1), we then point out a fundamental issue with this approach (subsection 2.2), and we finally propose a complementary approach and explain why it could address this issue (subsection 2.3).

To help us reason about the different approaches that continual learning methods could take, in this section we consider the following continual learning problem. Assume a model $f_w$, parameterized by $w$, that has learned a set of weights $\widehat{w}_{\text{old}}$ for an initial task[1], or a set of tasks, by optimizing a loss function $\ell_{\text{old}}$ on training data $D_{\text{old}} \sim \mathcal{D}_{\text{old}}$. We then wish to continue training the same model on a new task, by optimizing a loss function $\ell_{\text{new}}$ on training data $D_{\text{new}} \sim \mathcal{D}_{\text{new}}$, in such a way that the model maintains (or possibly improves) its performance on the previously learned task(s). As has been thoroughly described in the continual learning literature, if the model is trained on the new task in the standard way (i.e., optimize the new loss $\ell_{\text{new}}$ with stochastic gradient descent), the typical result is catastrophic forgetting and a solution $\widehat{w}_{\text{new}}$ that is good for the new task but no longer for the old one(s).

## 2.1 The Standard Approach to Continual Learning: Improving the Loss Function

To mitigate catastrophic forgetting, continual learning research from the past few years has typically focused on making changes to the loss function that is optimized. In particular, rather than optimizing the loss on the new task, many continual learning methods can be interpreted as optimizing an approximate version of the joint loss:

$$\widetilde{\ell}_{\text{joint}} = \ell_{\text{new}} + \widetilde{\ell}_{\text{old}} \qquad (1)$$

with $\widetilde{\ell}_{\text{old}}$ the method's proxy for the loss on the old tasks.

A straight-forward example of this approach is 'experience replay', which approximates $\ell_{\text{old}}$ by revisiting a subset of previously observed examples that are stored in an auxiliary memory buffer. In continual learning experiments, typically limits are imposed on the buffer's storage capacity and/or on the computational budget for training the model (Lesort et al., 2020; Wang et al., 2022; Prabhu et al., 2023). Both constraints prevent full replay of all previously observed data, meaning that $\ell_{\text{old}}$ can only be approximated. A wide range of studies aims to improve the quality of this approximation, for example by modifying the way samples

---

[1]The term 'task' is used here in a rather general way; it loosely refers to a combination of a data distribution and a loss function.

are selected to be stored in the buffer (Rebuffi et al., 2017; Chaudhry et al., 2019b; Aljundi et al., 2019b; Lin et al., 2021; Mundt et al., 2023), or by adaptively selecting which samples from the buffer to replay (Riemer et al., 2018; Aljundi et al., 2019a). As an alternative to storing past samples explicitly, generative models can be learned to approximate the input distributions of previous tasks (Robins, 1995; Shin et al., 2017; van de Ven et al., 2020).

Another popular class of methods for continual learning is based on regularization. As proxy for the loss on the old tasks, these methods add regularizing terms to the loss that impose penalties either for changes to the network's weights ('parameter regularization'; Kirkpatrick et al., 2017; Zenke et al., 2017; Aljundi et al., 2018) or for changes to the network's input-output mapping ('functional regularization'; Li and Hoiem, 2017; Dhar et al., 2019; Lee et al., 2019; Titsias et al., 2020). That these methods can be interpreted as attempts to approximate the joint loss can be shown by taking either a Bayesian perspective (Nguyen et al., 2018; Farquhar and Gal, 2019; Kao et al., 2021; Rudner et al., 2022) or a geometric perspective (Kolouri et al., 2020).

In summary, both replay- and regularization-based methods for continual learning operate by changing the loss function that is optimized, often with the aim of creating an approximate version of the joint loss. When developing new continual learning methods of this kind, the challenge is to design better objective functions.

## 2.2 The Stability Gap: A Challenge for the Standard Approach to Continual Learning

It was recently pointed out that even when replay- or regularization-based methods are considered to perform well (in the sense that they obtain good performance on both old and new tasks after finishing training on the new task), these methods still suffer from substantial, albeit often temporary, forgetting during the initial phase of training on a new task (De Lange et al., 2023). Until recently, this phenomenon – referred to as the *stability gap* – had not been observed, or been paid little attention to, due to the common evaluation setup in continual learning that only measures performance after training on the new data has converged. Nevertheless, the stability gap is undesirable, especially for safety-critical applications in which sudden drops in performance can be highly problematic. But also from the perspective of computational efficiency, or even if only the final learning outcome is of interest, avoiding the stability gap might be beneficial, as preventing forgetting seems easier and more efficient than having to re-learn later on (see Figure 4 of Van de Ven et al. (2020) for empirical support for this intuition).

Why does the stability gap happen? One possibility is that the stability gap is due to imprecision in the approximations

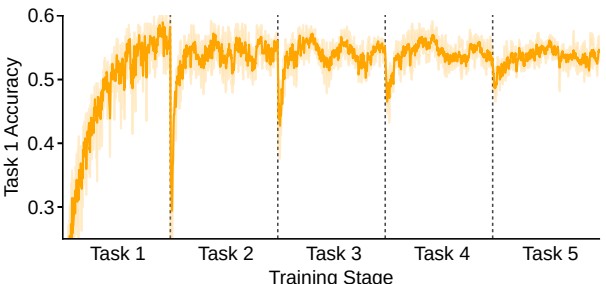

Figure 1: **The stability gap occurs even with incremental joint training (or 'full replay').** Shown is the test accuracy on the first task while the network is incrementally trained on all five tasks of Domain CIFAR. During the $n$-th task, the network is trained jointly on all training data from the first $n$ tasks. Even with this ideal approximation to $\ell_{\text{joint}}$, performance severely drops upon encountering a new task. Displayed are the means over five repetitions, shaded areas are $\pm 1$ standard error of the mean. Vertical dashed lines indicate task switches.

of the joint loss made by replay and regularization. If this were the case, it could be interpreted as good news for the standard approach to continual learning, as it would imply that by continuing to improve the quality of replay or regularization the stability gap could be overcome. However, this is not the case, as in preliminary experiments we find that the stability gap is consistently observed even with incremental joint training (Figure 1). This indicates that with better approximations to the joint loss alone, the stability gap cannot be solved.

### 2.3   Proposed Complementary Approach: Improving the Optimization Trajectory

The above observations relating to the stability gap suggest that the standard approach to continual learning of focusing on the loss function is not sufficient. We believe that continual learning would benefit from an additional perspective: rather than concentrating only on improving the optimization objective (i.e., what loss function to optimize), we argue that continual learning should also focus on improving the optimization trajectory (i.e., how to optimize that loss function).

To help explain why we believe that focusing on the optimization trajectory can yield benefits, we revisit the continual learning problem discussed at the start of section 2, which is schematically illustrated in Figure 2. Starting point is a model $f_w$ that has already learned a solution $\widehat{w}_{\text{old}}$ by optimizing loss $\ell_{\text{old}}$. Continuing to train this model by optimizing loss $\ell_{\text{new}}$ would result in catastrophic forgetting. Instead, as discussed, the standard approach in continual learning is to optimize $\widetilde{\ell}_{\text{joint}} = \ell_{\text{new}} + \widetilde{\ell}_{\text{old}}$, an approximation to the joint loss, rather than $\ell_{\text{new}}$. Optimizing a suit-

ably approximated version of the joint loss results in a solution $\widehat{w}_{\text{joint}}$ that is good for both the old and the new tasks. However, if this loss is optimized with standard stochastic gradient descent, the trajectory that is taken from $\widehat{w}_{\text{old}}$ to $\widehat{w}_{\text{joint}}$ goes through a region in parameter space where the loss on the old tasks is high. The corresponding transient drop in performance on the old tasks is the stability gap.

A first possibility that must be dealt with is that the stability gap is unavoidable, in the sense that there simply is no path from $\widehat{w}_{\text{old}}$ to $\widehat{w}_{\text{joint}}$ that does not traverse a region where the performance on old tasks is poor. Although this is theoretically possible, this option seems unlikely given recent work on mode connectivity in deep neural networks (Draxler et al., 2018; Garipov et al., 2018) showing that different local optima found by stochastic gradient descent are often connected by simple paths of non-increasing loss. In particular, Mirzadeh et al. (2021) showed that when optimizing a neural network using stochastic gradient descent on the joint loss while starting from a single task solution, the resulting joint solution is connected to the single task solution by a linear manifold of low loss on the single task. The same holds when, starting from a single task solution, the replay-approximated joint loss is optimized rather than the joint loss itself (Verwimp et al., 2021).

The above work on mode connectivity thus suggests that by changing the optimization routine it should be possible to avoid the stability gap. Besides that reducing the stability

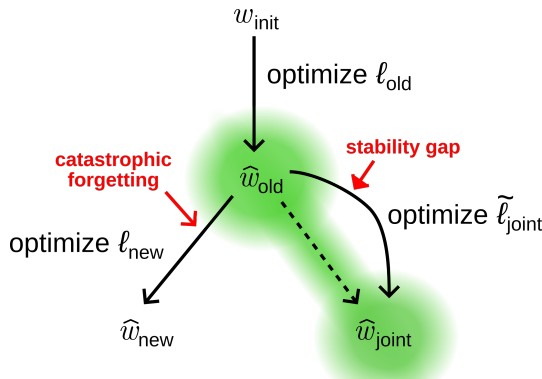

Figure 2: **Schematic of the stability gap, and how adjusting the optimization trajectory could avoid it.** When, starting from a solution for the old tasks ($\widehat{w}_{\text{old}}$), a proxy of the joint loss ($\widetilde{\ell}_{\text{joint}}$) is optimized with standard stochastic gradient descent, the optimization trajectory first passes through a region in parameter space with high loss on the old tasks before converging to a solution that is good for all tasks ($\widehat{w}_{\text{joint}}$). Work on mode connectivity suggests that a low-loss path between $\widehat{w}_{\text{old}}$ and $\widehat{w}_{\text{joint}}$ exists as well (dashed arrow), indicating that it should be possible to overcome the stability gap with a different optimization routine. Green shading indicates areas of low loss on the old tasks.

gap is important for safety-critical applications, we believe that it could bring other benefits for continual learning as well. For example, getting rid of the repeated re-learning cycles that characterize the stability gap could increase the learning efficiency. Moreover, if the loss function is non-convex, as is the case with deep neural networks, changing the way the loss is optimized could also lead to different, and hopefully better, final learning outcomes. The possibility of improved final performance is supported by the observation from Caccia et al. (2022) that the abrupt forgetting after task switches is not always recovered later on. This leads us to the following three hypotheses:

---

**Main hypothesis**
Better optimization routines for continual learning can:
 **(H1)** reduce the stability gap.

**Secondary hypotheses**
Reducing the stability gap can:
 **(H2)** increase learning efficiency;
 **(H3)** improve the final learning outcome.

---

**How to Improve Optimization for Continual Learning?**
In the above we have made an argument that continual learning should focus on improving its optimization routines, but we have not yet discussed *how* this could be done. To avoid the stability gap, an optimization routine is needed that is less greedy than standard stochastic gradient descent and that favors parameter updates that do not substantially increase the loss on old tasks. Interestingly, as we review in section 3, optimization routines based on gradient projection have been explored in the continual learning literature that already have these properties. Importantly, however, currently these gradient projection-based optimization routines are not used in the way envisioned by us, as they are used to optimize the loss on the new task rather than an approximated version of the joint loss.

## 2.4 Another Way to Avoid the Stability Gap?

An alternative approach that can circumvent the stability gap in continual learning is to use certain parts of the network only for specific tasks. This approach is employed by network expansion methods (Rusu et al., 2016; Yoon et al., 2018; Yan et al., 2021) and parameter isolation methods (Serra et al., 2018; Masse et al., 2018). To see why this approach can help to avoid the stability gap, consider the extreme case of using a separate sub-network for each task. In this case there is no forgetting at all, and thus also no stability gap. However, the use of task-specific components has some important disadvantages. Firstly, if task identity is not always provided, as is the case with domain- and class-incremental learning (van de Ven et al., 2022), it might not be clear which parts of the network should be used. This issue could be addressed by inferring task identity, for exam-

ple using generative models or other out-of-distribution detection techniques (van de Ven et al., 2021; Henning et al., 2021; Kim et al., 2022; Zając et al., 2023), but such task inference can be challenging. Secondly, having separate parts of the network per task limits the potential of positive transfer between tasks, which is an important desideratum for continual learning (Hadsell et al., 2020).

# 3 GRADIENT PROJECTION-BASED OPTIMIZATION

A tool that has been explored in the continual learning literature for modifying the way a given loss function $\ell(w)$ is optimized is 'gradient projection'. With gradient projection, rather than basing the parameter updates on the original gradient $g = \nabla_w \ell(w)$, they are based on a projected version $\bar{g}$ of that gradient. Important from the perspective of our paper, gradient projection does not alter the loss function that is optimized (e.g., the loss landscape and its local minima remain unchanged), it only changes the way the loss function is optimized (Kao et al., 2021). In this section we review two current lines of continual learning studies that make use of gradient projection.

## 3.1 Orthogonal Gradient Projection

Orthogonal gradient projection methods aim to avoid interference between tasks by confining the training of each new task to previously unused subspaces. To restrict parameter updates to directions that do not interfere with the performance on old tasks, the gradient of the loss on the new task is projected to the orthogonal complement of the 'gradient subspaces' of old tasks. Various ways to construct these gradient subspaces have been proposed. Several studies use the subspaces spanned by all layer-wise inputs of old tasks, which they characterize using conceptors (He and Jaeger, 2018) or by iteratively accumulating projector matrices using a recursive least squares algorithm (Zeng et al., 2019; Guo et al., 2022). To reduce the memory and computational costs, Saha et al. (2021) approximate the input subspaces of each task using singular value decomposition and its $k$-rank approximation. Farajtabar et al. (2020) instead use the span of a set of stored gradient directions of old tasks as the gradient subspace to protect.

Constraining sequential optimization to orthogonal subspaces can mitigate forgetting effectively, but it restricts the learning of new tasks to successively smaller subspaces and it eliminates the potential to further improve the model with respect to old tasks. Orthogonal gradient projection methods especially struggle or break down when the input spaces of different tasks substantially overlap with each other (He, 2018).

Recent advancements in this line of work therefore focus on relaxing the constraints of the orthogonal projection

framework to enable knowledge transfer between tasks. Deng et al. (2021) dynamically scale gradients and search for flatter minima, and Lin et al. (2022) use the idea of 'trust regions' (Schulman et al., 2015) to selectively relax constraints for protected subspaces of old tasks most related to the new task. As an alternative, Kao et al. (2021) bring a Bayesian perspective to gradient projection. They use the inverse of a Kronecker-factored approximation to the Fisher information matrix as their projector matrix, and additionally protect previous knowledge by parameter-based regularization.

### 3.2 Gradient Episodic Memories

Another class of gradient projection-based optimization methods for continual learning, which also does not enforce strict orthogonality of future updates, is based on Gradient Episodic Memory (GEM; Lopez-Paz and Ranzato, 2017). The projection mechanism of this approach is motivated by a constrained optimization problem, where the goal is to optimize $\ell_{\text{new}}$ without increasing $\ell_{\text{old}}$:

$$\min_w \ell_{\text{new}}(w), \ \text{ such that } \ell_{\text{old}}(w) \leq \ell_{\text{old}}(\widehat{w}_{\text{old}}) \quad (2)$$

To determine whether a parameter update based on $g = \nabla_w \ell_{\text{new}}(w)$ might increase $\ell_{\text{old}}$, the gradient(s) for the old task(s) are estimated using examples from a replay buffer: $g_{\text{old}} = \nabla_w \widetilde{\ell}_{\text{old}}(w)$. If the directions of $g$ and $g_{\text{old}}$ align (in the sense that their angle does not exceed $90°$), it is conjectured that a parameter update based on $g$ is unlikely to increase $\ell_{\text{old}}$, and $g$ is left unchanged. If the angle between $g$ and $g_{\text{old}}$ exceeds $90°$, $g$ is projected to $\bar{g} = g - \frac{g^{\text{T}} g_{\text{old}}}{g_{\text{old}}^{\text{T}} g_{\text{old}}} g_{\text{old}}$, which is the closest gradient to $g$ (in $l_2$-norm) with a $90°$ angle to $g_{\text{old}}$. Because in our continual learning example it is the case that $\ell_{\text{old}}$ encompasses all past tasks, this description actually corresponds to Averaged GEM (A-GEM; Chaudhry et al., 2019a), a computationally more efficient version of GEM. The original formulation of GEM enforces $\bar{g}$ to align with the gradient of each individual past task (Lopez-Paz and Ranzato, 2017).

Given that GEM and A-GEM explicitly aim to prevent increases of the loss on old tasks, these methods might be able to avoid the stability gap. Empirically, however, this is not the case, as GEM suffers from considerably larger stability gaps than experience replay (De Lange et al., 2023). Moreover, also in terms of final performance, experience replay consistently outperforms both GEM and A-GEM (De Lange et al., 2022; van de Ven et al., 2022).

We expect that the disappointing performance of GEM is due to its choice of objective function: GEM optimizes the loss on the new task (i.e., $\ell_{\text{new}}$) rather than an approximation to the joint loss (i.e., $\widetilde{\ell}_{\text{joint}}$). In other words, we believe that GEM under-utilizes its replay buffer by solely delineating gradient constraints but not actively optimizing the replay-approximated joint loss. When GEM was proposed,

it was assumed that directly optimizing a joint loss approximated with a relatively small replay buffer could not work well due to overfitting (Lopez-Paz and Ranzato, 2017), but recent work indicates such overfitting is not as detrimental as thought (Chaudhry et al., 2019b; Verwimp et al., 2021). Nevertheless, as far as we are aware, changing GEM's objective function has not been explored.

## 4 PROOF-OF-CONCEPT EXPERIMENTS

As discussed in the last section, the gradient projection-based optimization routine of GEM encourages parameter updates that do not strongly interfere with old tasks without imposing overly strict constraints that would fully segregate tasks. Yet, so far this optimization routine has not been used to optimize proxies of the joint loss. This makes GEM a convenient tool for a first set of proof-of-concept experiments to evaluate the merits of our proposition that continual learning should consider both *what* and *how* to optimize. We plan to combine GEM's optimization routine both with a basic version of experience replay that explicitly approximates the joint loss (subsection 4.1) and with state-of-the-art replay-based methods (subsection 4.2).

### 4.1 Experience Replay with Gradient Projection-based Optimization

In a first set of experiments we test whether, when the optimization objective is a standard replay-approximated version of the joint loss, using GEM's gradient projection-based optimization routine provides the benefits hypothesized in subsection 2.3.

**Approximating the Joint Loss** To approximate the joint loss we use a basic version of Experience Replay (ER). In our implementation of ER, at the end of each task new examples are added to the memory buffer using class-balanced sampling from the training set, and in each training iteration uniform sampling from the buffer is used to choose which samples to replay. To approximate the joint loss as closely as possible, when training on the $n$-th task, we balance the loss on the current data and the loss on the replayed data using $\widetilde{\ell}_{\text{joint}} = \frac{1}{n}\ell_{\text{new}} + (1 - \frac{1}{n})\widetilde{\ell}_{\text{old}}$. In each iteration, the total number of replayed samples from all past tasks combined is always equal to $b$, which is the size of the mini-batch from the current task.

In addition to approximating the joint loss with ER, we also run experiments using the joint loss itself. For this, all training data from past tasks are stored, and in each iteration we use $b$ samples from each past task to compute $\ell_{\text{old}}$. This can be thought of as 'full replay'.

**Optimization Trajectory** To improve the sub-optimal optimization trajectory that is taken by standard ER, we use the gradient projection-based optimization routines of

GEM and A-GEM. Importantly, we only use the optimization routines of GEM and A-GEM, not their optimization objectives. As optimization objective we instead use the replay-approximated joint loss $\widetilde{\ell}_{\text{joint}}$. To achieve this, in the description of GEM in the first paragraph of subsection 3.2, we only need to replace all mentions of $\ell_{\text{new}}$ with $\widetilde{\ell}_{\text{joint}}$. To further illustrate our proposed combination approach, pseudocode for ER + A-GEM is provided in Algorithm 1.

---

**Algorithm 1** ER + A-GEM

---

**Require:** parameters $w$, loss function $\ell$, learning rate $\lambda$, data stream $\{D_1, ..., D_T\}$

$M \leftarrow \{\}$
**for** $t = 1, ..., T$ **do**
    **for** $(x, y) \in D_t$ **do**
        $g \leftarrow \nabla_w \ell(f_w(x), y)$
        $(\tilde{x}, \tilde{y}) \leftarrow \text{SAMPLE}(M)$
        $g_{\text{old}} \leftarrow \nabla_w \ell(f_w(\tilde{x}), \tilde{y})$
        $g_{\text{joint}} \leftarrow \frac{1}{t} g + (1 - \frac{1}{t}) g_{\text{old}}$
        $\bar{g} \leftarrow \text{PROJECT}(g_{\text{joint}}, g_{\text{old}})$
        $w \leftarrow \text{OPTIMIZER\_STEP}(w, \lambda, \bar{g})$
    **end for**
    $M \leftarrow \text{UPDATE\_BUFFER}(M, D_t)$
**end for**

---

**function** PROJECT($g, g_{\text{ref}}$)
    **if** $g^{\text{T}} g_{\text{ref}} \geq 0$ **then**
        **return** $g$
    **else**
        **return** $g - \frac{g^{\text{T}} g_{\text{ref}}}{g_{\text{ref}}^{\text{T}} g_{\text{ref}}} g_{\text{ref}}$
    **end if**
**end function**

---

**Approaches to Compare** The main experimental comparison of interest is between standard ER and our proposed combination approach ER + GEM, as this allows testing whether, when doing continual learning by optimizing a proxy of the joint loss, benefits can be gained by changing the way this objective is optimized. Additionally, to probe the individual contributions of the optimization objective and the optimization routine, we also include GEM itself and continual finetuning in our experimental comparison. See Table 1 for an overview of the approaches we compare. In this table ER can be replaced by 'full replay', and GEM can be replaced by A-GEM. We run experiments with all combinations of these base methods.

### 4.2 Improving State-of-the-art

Next we ask whether the use of gradient projection-based optimization could improve the performance of state-of-the-art replay-based methods. To test this, we run the methods Dark Experience Replay (DER; Buzzega et al., 2020) and Bias Correction (BiC; Wu et al., 2019) both with and

Table 1: Overview of the approaches to compare in our proof-of-concept experiment, illustrated with ER and GEM as base methods. GP: gradient projection.

| Method | Approximate joint loss | GP-based optimization |
|---|---|---|
| Finetuning | ✗ | ✗ |
| ER | ✓ | ✗ |
| GEM | ✗ | ✓ |
| ER + GEM | ✓ | ✓ |

without using the optimization routine of A-GEM. For these experiments we only consider A-GEM, as it is not straight-forward to combine DER with the original version of GEM. We implement DER and BiC according to their original papers. For completeness, details for both methods will be provided in the Appendix. As BiC is a method that is specialized for class-incremental learning, it is included only with the class-incremental learning benchmarks.

## 5 EXPERIMENTAL PROTOCOL

### 5.1 Setup

We consider a task-aware supervised continual learning setting, with a task sequence $\mathcal{T} = \{T_1, ..., T_T\}$ of $T$ disjoint classification tasks $T_t$. A fixed capacity neural network model $f_w$ is incrementally trained on these tasks with a cross-entropy classification loss. When training on task $T_t$, the model has only access to the training data $D_t = \{X_t, Y_t\}$ of that task and the data in the memory buffer (see below), and the goal is to learn a model with strong performance on all tasks $T_{\leq t}$ encountered so far. The model may be evaluated after any parameter update.

**Benchmarks** We conduct our study on four benchmarks, covering the domain- and class-incremental learning scenarios (van de Ven et al., 2022). As class-incremental learning benchmarks we use Split CIFAR-100, which is based on the CIFAR-100 dataset (Krizhevsky et al., 2009), and Split Mini-Imagenet, which is based on Mini-Imagenet (Vinyals et al., 2016). Both original datasets contain 50,000 RGB images of 100 classes; CIFAR-100 in resolution 32x32, Mini-Imagenet in resolution 84x84. For Split CIFAR-100 the classes are divided into ten tasks with ten classes each, for Split Mini-Imagenet the classes are divided into twenty tasks with five classes each. In both cases, the classes are divided over the tasks randomly, and for each random seed a different division is used. We also use the CIFAR-100 dataset to construct Domain CIFAR, a domain-incremental learning benchmark. For this benchmark, each of the twenty super-classes of CIFAR-100 is split across five tasks, such that every task contains

one member of each super-class (i.e., there are twenty classes per task, one from each super-class). The goal in each task is to predict to which super-class a sample belongs. The other domain-incremental learning benchmark is Rotated MNIST. Each task consists of the entire MNIST dataset (LeCun et al., 1998) with a certain static rotation applied. We construct three tasks with rotations $\{0°, 80°, 160°\}$, as De Lange et al. (2023) found these to provoke the largest stability gaps without inducing ambiguity between digits 6 and 9 (which would happen with rotations close to $180°$).

**Architectures** For the Rotated MNIST benchmark, we use a fully-connected neural network with two hidden layers of 400 ReLUs each, followed by a softmax output layer. For the other benchmarks, following Lopez-Paz and Ranzato (2017), we use a reduced ResNet-18 architecture. Compared to a standard ResNet-18 (He et al., 2016), this architecture has three times less channels in each layer and replaces the $7 \times 7$ kernel with stride of 2 in the initial convolutional layer by a $3 \times 3$ kernel with stride of 1. The latter prevents an early stark information reduction for images with small resolution. All benchmarks are trained with a single-headed final layer that is shared between all tasks.

**Memory Buffer** The memory buffer can store up to 100 samples of each class. For the domain-incremental learning benchmarks this means 100 samples of each class per task (e.g., with Rotated MNIST, for each digit the buffer can store 100 examples with rotation $0°$, 100 examples with rotation $80°$ and 100 examples with rotation $160°$). Exceptions to this are the experiments with 'full replay', in which all training data are stored.

**Offline & Online** All experiments are run in both an 'offline version' and an 'online version'. In the offline version, multiple passes over the data are allowed, and the number of training iterations is set relatively high to encourage near convergence for each task. In the online version only a single epoch per task is allowed (i.e., each sample is seen just once, with the exception if it is replayed from memory).

**Training Hyperparameters** All models are trained using an SGD optimizer with momentum 0.9 and no weight-decay. When gradient projection is used, this optimizer acts on the projected gradients. Except for whitening (with mean and standard deviation of the respective full training sets), no data augmentations are used. Exceptions to this are the experiments with DER and BiC, for which we use the data augmentations described in the original papers that proposed these methods (Buzzega et al., 2020; Wu et al., 2019). In the offline experiments, we train with mini-batch size 128 for around five epochs (Rotated MNIST) or ten epochs (Domain CIFAR, Split CIFAR-100 and Split Mini-Imagenet) per task. To be exact, we use 2000 iterations

per task for Rotated MNIST, 800 for Domain CIFAR, 400 for Split CIFAR-100, and 200 for Split Mini-Imagenet. For each experiment in the offline setting, we sweep a set of static learning rates $\{0.1, 0.01, 0.001\}$. For each experiment in the online setting, we sweep both a set of mini-batch sizes $\{10, 64, 128\}$ and a set of static learning rates $\{0.1, 0.01, 0.001\}$. In the online setting, the number of iterations per task is determined by the selected mini-batch size and the number of training samples.

## 5.2 Evaluation

We track the performance of all methods throughout training using 'continual evaluation' (De Lange et al., 2023). In particular, after every training iteration we evaluate for each task the accuracy of the model on a hold-out test set.

**Testing the Hypotheses** To quantitatively compare the stability gap of different approaches (i.e., to evaluate **H1**), we use the 'average minimum accuracy' metric defined by De Lange et al. (2023). For completeness, details of this metric will be provided in the Appendix. To qualitatively compare the stability gaps, we plot per-task accuracy curves with per-iteration resolution (e.g., as in Figure 1). To compare the learning efficiency of different approaches (i.e., to evaluate **H2**), we use the final average accuracy of the online experiments. To compare the final learning outcomes of different approaches (i.e., to evaluate **H3**), we use the final average accuracy of the offline experiments.

**Computational Complexity** To provide insight into the computational complexity of the considered methods, we report for each method its empirical training time on the online version of Split CIFAR-100. For this evaluation all methods are run on identical hardware.

**Standard Errors** Each experiment is run five times, with a different random seed and different division of the classes over tasks for each run. For each metric, both the mean over these runs and the standard error of the mean are reported.

## 6 OUTLOOK

Motivated by the stability gap in replay- and regularization-based methods, in this pre-registered report we proposed that strategies for continual learning should focus not only on *what* to optimize, but also on *how*. To empirically evaluate the merits of this conceptual proposition, we described a set of proof-of-concept experiments in which gradient projection-based optimization is used to optimize replay-approximated versions of the joint objective. We will now perform these experiments by rigorously following the detailed protocol laid out in this report. We expect to communicate the results of these experiments at the latest in June 2024.

## Acknowledgements

We thank Matthias De Lange, Eli Verwimp and anonymous reviewers for useful comments. This project has been supported by funding from the European Union under the Horizon 2020 research and innovation program (ERC project KeepOnLearning, grant agreement No. 101021347) and under Horizon Europe (Marie Skłodowska-Curie fellowship, grant agreement No. 101067759).

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
