# OpenReview forum: "Two Complementary Perspectives to Continual Learning: Ask Not Only What to Optimize, But Also How"
_continualai.org/CLAI/2023/Unconference_Preregistration_Track — 1st CLAI Unconf_

### Official Review · Reviewer_cMwy · 2023-08-04
**Borderline contribution**

**Clarity:** 3
**Originality:** 2
**Soundness:** 3
**Significance:** 2
**Rating:** 5
**Confidence:** 4

**Review:**

This paper proposed an approach based on the combination of Experience Replay and Gradient Projection Optimization (based on GEM). The aim is reducing the stability gap at the beginning of each new task, and possibly improving the learning efficacy.

**Strengths:**

The idea proposed in not particularly novel (or high risk).

**Weaknesses:**

The experimental setup proposed to validate the contribution must be improved/extended (see below in Protocol)




**Questions:**

See Protocol

**Protocol:**

-	Other baseline approaches such as DER and BIC need to consider for comparison. In fact, neither GEM nor ER are state-of-the-art and demonstrating small advantages over them is not particularly interesting

-	The computation complexity must be taken into account by comparing the complexity (e.g., training time) of all the methods considered. In fact, a small accuracy advantage could be useless in case of a large efficiency drop.

---

### Official Review · Reviewer_DPNg · 2023-08-17
**Interesting topic, but limited technical novelty and impact**

**Clarity:** 2
**Originality:** 2
**Soundness:** 3
**Significance:** 2
**Rating:** 5
**Confidence:** 4

**Review:**

This paper addresses the stability gap in continual learning by proposing optimizing a balanced loss and a gradient projection strategy to adjust the optimization trajectory. The proposed method, ERGO, is expected to reduce the stability gap, improve learning efficiency, and final performance.

**Strengths:**

The stability gap is an interesting in continual learning and requires deeper understandings and advances methods to tackle this challenge.

**Weaknesses:**

- My most important concern of this work that the scope is quite board, making it difficult to connect the contributions to demonstrate a strong impact. Particularly, the authors hypothesized  that the proposed method could: (i) reduce the stability gap; (ii) increase learning efficiency; and (iii) improve the final learning outcome, which are three major challenges of continual learning. However it is quite unclear if there are any correlations between the stability gap and the learning efficiency or outcome. Thus, I feel the contribution of this work is quite spread out and does not have a strong, significant impact to any of the three challenges.

- Second, the proposed method is quite poorly described. The authors spent a significant amount of contents (4 pages) to cover the basics of continual learning, most of which are considered standard knowledge. However, the are very little efforts in detailing exactly the proposed method's procedure. It would be helpful to provide a pseudo-code of the proposed algorithm. Based on the existing proposal, the proposed method seems to be a trivial combination of some existing continual learning techniques, which limits its technical contribution.

**Questions:**

- How addressing the stability gap could improve the continual learning efficiency and overall performance?

- Key contribution of the proposed method compared to existing studies?


**Protocol:**

The evaluation protocol for learning efficiency might be problematic. While it make sense for the online continual learning setting, I think it is not suitable for the offline setting. Measuring the learning efficiency has been studied in the meta learning literature and the authors could adopt/adapt the metric used in OML [A].

[A] Finn, Chelsea, et al. "Online meta-learning." International Conference on Machine Learning. PMLR, 2019.

---

### Official Review · Reviewer_C96a · 2023-08-19
**The idea is not well developed yet. The direction is fine, but some more work is needed.**

**Clarity:** 3
**Originality:** 3
**Soundness:** 2
**Significance:** 2
**Rating:** 6
**Confidence:** 4

**Review:**

Too much discussion about previous approaches. The new idea is not well developed. The prototype implementation has used some existing methods.

**Strengths:**

(1). I think the idea of working on alternative optimization objective and trajectory is an interesting one. I wonder whether the phenomenon in Figure 1 can be exploited to design an effective method.
(2) The evaluation protocol is fine.
(3) The paper is easy to read.

**Weaknesses:**

(1). It is not clear whether you want to improve the gradient based approach as it isn't very new. As the replay data is not so much, optimizing the joint objection will not be able to solve the forgetting problem.

(2). In Section 2, you may also want to include the gradient orthogonal projection approaches (which you mentioned later), network expansion approaches (e.g. DER: Dynamically expandable representation for class incremental learning. CVPR-2021) and the OOD based approaches (e.g., "A Theoretical Study on Solving Continual Learning," NeurIPS-2022).

(3). Writing is wordy.

**Questions:**

No.

**Protocol:**

It is fine.

---

### Official Review · Reviewer_zAqm · 2023-08-20

**Clarity:** 2
**Originality:** 1
**Soundness:** 2
**Significance:** 1
**Rating:** 4
**Confidence:** 4

**Review:**

The soundness of the paper is commendable; however, its overall completeness is lacking. While the experimental settings are adequately described in the paper, the actual results of these experiments are not reported. The methodology itself can be seen as an enhanced version of GEM. Nevertheless, the intriguing aspect lies in the notion of modifying the optimization routine with a replay-approximated joint loss. The incompleteness of the paper's content stands as the primary reason for its rejection.

**Strengths:**

1. the notion of modifying the optimization routine with a replay-approximated joint loss is interesting

**Weaknesses:**

The incompleteness of the paper's content

**Questions:**

The absence of experimental results is apparent; however, if I have missed them, I kindly request clarification.

**Protocol:**

From an evaluation protocol perspective, considering domain- and class-incremental learning benchmarks is deemed sufficient.

---

### Decision · Program_Chairs · 2023-09-12

**Decision:**

Accept

**Comment:**

Dear authors,

Congratulations, your paper has been accepted at the ContinualAI Unconference 2023! We look forward to engaging in further discussions with you and others in the community.

Details will follow shortly regarding camera-ready versions. Please do take the feedback from reviews into account.

Although reviewers mostly agreed that idea of optimizing both the objective and the trajectory is interesting, there were concerns over how interesting and novel the proposed method (combining ER and GEM) is. For this Unconference, we want to focus on the motivation and hypotheses, provided they are backed by detailed experimental protocol. I think the authors could better motivate why combining the proposed methods specifically helps solve the motivated problem / improve the link between the hypotheses and the proposed methods.
- Does GEM specifically cause optimization trajectories to follow the low-loss path between modes? Why/why not? The authors could consider having a toy case in the camera-ready version where they show that GEM optimizes (or does not optimize) in this way.
- The authors said they will test more objectives (DER and BiC) instead of just ER.
- The authors agreed to lessen the claims regarding two of the previously-three hypotheses, as these two are not as well-motivated in the paper.